# LIGHTCTRL: TRAINING-FREE CONTROLLABLE VIDEO RELIGHTING

**Yizuo Peng**[1,2]    **Xuelin Chen**[3],   **Kai Zhang**[1,*],   **Xiaodong Cun**[2,*]
[1] Tsinghua University,    [2] GVC Lab, Great Bay University,    [3] Adobe Research
pengyz23@mails.tsinghua.edu.cn,cun@gbu.edu.cn

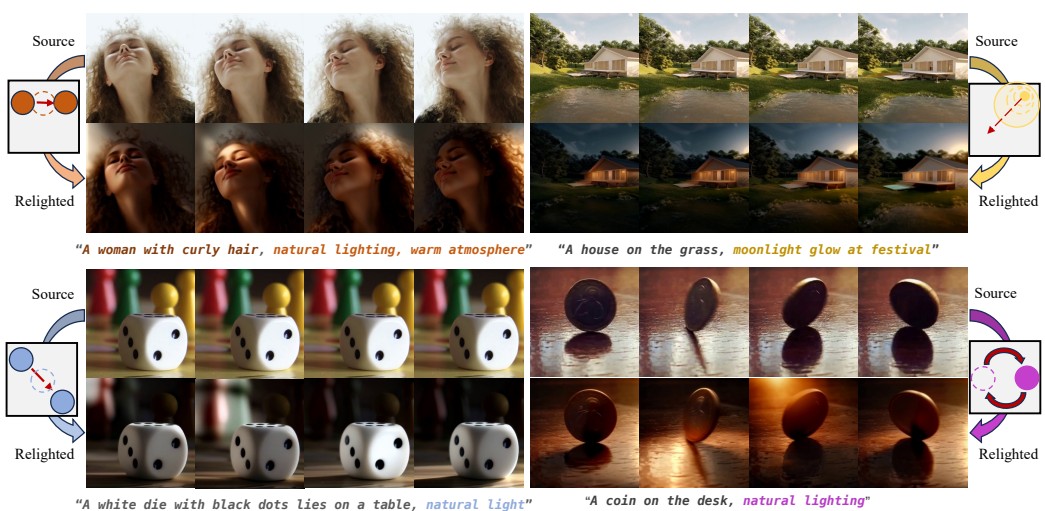

Figure 1: LightCtrl can relight an input video to produce high-quality results with strong temporal consistency and, particularly, illumination that closely follows the user-specified light trajectory.

## ABSTRACT

Recent diffusion models have achieved remarkable success in image relighting, and this success has quickly been reproduced in video relighting. Although these methods can relight videos under various conditions, their ability to explicitly control the illumination in the relighted video remains limited. Therefore, we present LightCtrl, the first controllable video relighting method that offers explicit control over the video illumination through a user-supplied light trajectory in a training-free manner. This is essentially achieved by leveraging a hybrid approach that combines pre-trained diffusion models: a pre-trained image relighting diffusion model is used to relight each frame individually, followed by a video diffusion prior that enhances the temporal consistency of the relighted sequence. In particular, to enable explicit control over dynamically varying lighting in the relighted video, we introduce two key components. First, the Light Map Injection module samples light trajectory-specific noise and injects it into the latent representation of the source video, significantly enhancing illumination coherence with respect to the conditional light trajectory. Second, the Geometry-Aware Relighting module dynamically combines RGB and normal map latents in the frequency domain to suppress the influence of the original lighting in the input video, thereby further improving the relighted video's adherence to the input light trajectory. Our experiments demonstrate that LightCtrl can generate high-quality video results with diverse illumination changes closely following the light trajectory condition, indicating improved controllability over baseline methods. The code will be released at: https://github.com/GVCLab/LightCtrl.

---

*Corresponding authors

# 1 INTRODUCTION

The ability to controllably edit illumination in images and videos is a highly desired feature in visual content creation. With the rapid advancement of generative models, image relighting has achieved remarkable success. These image relighting pipelines (Sun et al., 2019; Nestmeyer et al., 2020; Pandey et al., 2021; Zeng et al., 2024; Bharadwaj et al., 2024; He et al., 2024; Ponglertnapakorn et al., 2023; Ren et al., 2024; Kim et al., 2024) typically involve a complex traditional synthesis pipeline that adheres to the desired relighting conditions. More recently, the state-of-the-art image relighting model IC-Light (Zhang et al., 2025) directly fine-tunes a pre-trained diffusion-based image generator conditioned on various input lighting scenarios.

The success is quickly reproduced in video relighting, where, analogically, video diffusion models (VDMs) are repurposed for relighting existing videos. RelightVid (Fang et al., 2025) constructs a high-quality video dataset featuring diverse lighting conditions to train the VDM. Nevertheless, the data crafting is often prohibitively expensive. This limitation has prompted the development of training-free methods, such as Light-A-Video (Zhou et al., 2025), which progressively integrates individually relighted frames, produced by a pre-trained image relighting model (Zhang et al., 2025), into the denoising process of a VDM.

These existing approaches (Fang et al., 2025; Zhou et al., 2025) often overlook the subtle yet crucial impact of localized light adjustments on the narrative and emotional expression within videos. We argue that a specific local light change in video frames can significantly enhance the storytelling atmosphere, as it allows for more nuanced manipulation of visual focus and mood. For instance, a gradual dimming of light in a particular scene corner can emphasize a character's inner tension, while a sudden beam of light can highlight a pivotal plot moment. To address this gap and mock the dynamic changes of light and shadow in videos, we define a controllable video relighting task. This task aims to provide fine-grained control over frame-wise lighting, enabling creators to tailor light effects to the narrative flow of each scene. By introducing procedural synthesis techniques, we seek to achieve precise, frame-by-frame light control, which can simulate real-world lighting dynamics such as the slow transition of dawn or the specific highlight region.

We thus present LightCtrl, the *first* controllable video relighting method that produces high-quality results with explicit control over the video illumination through a user-supplied light trajectory. Similar to previous work (Zhou et al., 2025; Meng et al., 2021), LightCtrl is based on the video editing pipeline and adopts a hybrid approach that combines pre-trained diffusion models: a pre-trained image relighting diffusion model is used to relight each frame individually, followed by a video diffusion prior that enhances the temporal consistency of the relighted sequence. In particular, to achieve controllability of video relighting through user-defined light trajectory, a Light Map Injection module is devised. Specifically, we begin by synthesizing a sequence of light maps that align with the lighting trajectory defined by the user. This can be straightforwardly achieved through procedural synthesis, or users have the flexibility to manually author the light map sequence. Then, we inject these light maps as control signals into the noise latent of both the single-frame relighting model and the video diffusion model. The Light Map Injection module significantly improves the adherence of the lighting in the generated video with respect to the input light trajectory. On the other hand, the illumination in the source video could greatly affect the relighting results, hence we introduce the Geometry-Aware Relighting module, which takes as input the estimated normal map sequence of the source video, to suppress the influence of the source illumination. Additionally, to preserve the details in the source video, we dynamically fuse the RGB video and normal maps latents in the frequency domain, obtaining the best of both worlds. Experiments demonstrate that our training-free method enables coherent and controllable video relighting across a wide range of source videos, faithfully following diverse lighting trajectories specified by the user.

# 2 RELATED WORK

**Illumination-Controlled Visual Generation.** Controllable illumination synthesis has become a crucial research direction in visual generation. Early efforts employed deep neural networks to control illumination on light stage data (Sun et al., 2019) and enhanced neural network relighting by incorporating physical priors (Nestmeyer et al., 2020). Total Relighting (Pandey et al., 2021) optimized the Phong model using HDR lighting maps. With the advent of diffusion-based models,

methods such as DiLightNet (Zeng et al., 2024), GenLit (Bharadwaj et al., 2024), DifFRelight (He et al., 2024), and DiFaReli (Ponglertnapakorn et al., 2023) have emerged, achieving detailed relighting. However, applying image-based methods such as Relightful Harmonization (Ren et al., 2024), SwitchLight (Kim et al., 2024), and IC-Light (Zhang et al., 2025) to videos often results in temporal inconsistency. In video illumination control, methods like EdgeRelight360 (Lin et al., 2024) and ReliTalk (Qiu et al., 2024a) addressed temporal consistency using 3D-aware models. Generative portrait-relighting works such as Lumos (Yeh et al., 2022) and related methods (Zhang et al., 2021; Cai et al., 2024) also achieve impressive relighting results by injecting illumination through environment maps. However, their strong dependence on environment-map inputs limits practical applicability. Training-based methods like RelightVid (Fang et al., 2025) achieve coherent video relighting through high-quality datasets, while training-free methods like Light-A-Video (Zhou et al., 2025) leverage image relighting techniques and video diffusion models' motion priors. Despite these advances, no existing method can yet achieve flexible video relighting based on user-defined lighting trajectories.

**Video Editing and Video Diffusion Models.** Recent years have witnessed substantial advancements in video generation, particularly through the application of video diffusion models (Blattmann et al., 2023; Guo et al., 2023; Yang et al., 2024; Xing et al., 2024), which are capable of producing high-quality, physically plausible videos. In the domain of text-to-video generation, various strategies have been explored: some methods integrate motion modeling modules into existing text-to-image (T2I) architectures (Guo et al., 2023; Chen et al., 2023; 2024; Wang et al., 2023a) to capture temporal dynamics, while others focus on learning video-specific priors from scratch (Yang et al., 2024). These innovations have also spurred interest in video editing techniques, where some approaches optimize models through additional training (Mou et al., 2024; Cheng et al., 2023), while others achieve editing without requiring further training (Ku et al., 2024; Ling et al., 2024). Additionally, when adapting image-based methods to video, researchers have employed different strategies, such as leveraging pre-trained T2I networks (Ma et al., 2024b; Wang et al., 2023b; Wu et al., 2023) or utilizing pre-trained optical flow models (Yang et al., 2023; Cong et al., 2023) to enhance temporal coherence. However, despite these notable achievements, precise control over video lighting remains a significant challenge, with existing methods falling short in maintaining lighting motion consistency, preserving scene details, and ensuring the rationality of character lighting.

**Controllable Video Generation.** With the rapid development of video generation models, significant potential has been brought to the field of controllable video generation. Recently, some methods have used basic conditions such as depths and sketches to control video content generation (Chen et al., 2024; Wang et al., 2023c). There are also methods based on motion (Geng & Owens, 2024), optical flows (Zhao et al., 2023), and those related to human poses (Ma et al., 2024b). Another category of methods (Wang et al., 2024a;b) provides user-defined bounding boxes or trajectories to achieve flexible control over the foreground content in videos. The emergence of most of these methods is largely due to the appearance of the ControlNet (Zhang et al., 2023) architecture. However, these methods often require us to construct high-quality training data to train the corresponding modules to achieve controllable generation. Accordingly, some training-free methods have emerged, such as Trailblazer (Ma et al., 2024a), which enhances attention on specific regions via cross-attention, Peekaboo (Jain et al., 2024), which uses masked attention for control without additional training, and FreeTraj (Qiu et al., 2024b), which aligns trajectories with given boxes using frequency fusion.

## 3 METHOD

The proposed LightCtrl is a training-free controllable video relighting method that provides controllability of the video illumination through the given light map conditioning. We begin by introducing the background of the image relighting method, , IC-Light (Zhang et al., 2025) in Sec. 3.1. Then we elaborate the overall pipeline of LightCtrl in Sec. 3.2, along with its two key components: Light Map Injection module (Sec. 3.2.2) and Geometric-Aware Relighting module (Sec. 3.2.3).

### 3.1 PRELIMINARY: IC-LIGHT (ZHANG ET AL., 2025) FOR SINGLE IMAGE RELIGHTING

IC-Light (Zhang et al., 2025) is the state-of-the-art method for conditional image relighting, which is fine-tuned based on a pre-trained text-to-image diffusion model, Stable Diffusion (Rombach et al.,

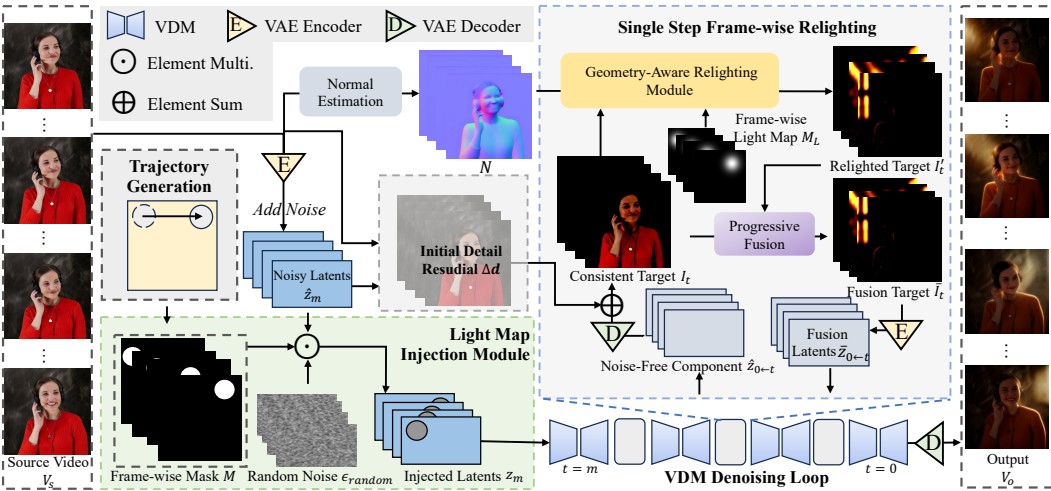

Figure 2: **Overall Pipeline.** We perform controllable video relighting with a user-provided light trajectory. Where we inject the light map on the noisy latent of VDM using the light map injection module. Then, in each denosing step, we design a geometry-aware relighting module to produce the relighted results frame-wise. Thus, the VDM can help to generate consistent video results with controllable lighting.

2021). During training, the target relighted image $I_L$ is encoded into a latent space via a pre-trained VAE encoder $\mathcal{E}$, where the noise is added iteratively via diffusion model theory (Ho et al., 2020). Given the set of conditions, including time step $t$, background reference $L$, and the input degradation $I_d$, the network $\delta$ learn to predict the added noise $\epsilon$ of each step. Besides the original diffusion progress, their key innovation is the light transport consistency constraint, which enforces the linear properties of light transport by minimizing the difference between the appearance under mixed illumination and the sum of appearances under individual illuminations, so that IC-Light can be trained via:

$$\mathcal{L} = \lambda_{\text{diffusion}}\|\epsilon - \delta(\mathcal{E}(I_L)_t, t, L, \mathcal{E}(I_d))\|_2^2 + \lambda_{\text{consistency}}\|M \odot (\epsilon_{L_1+L_2} - \phi(\epsilon_{L_1}, \epsilon_{L_2}))\|_2^2, \quad (1)$$

where $M$ is the foreground mask, $\odot$ denotes pixel-wise multiplication, and $\phi(\cdot, \cdot)$ is a learnable function. The final loss function is a weighted combination of the vanilla diffusion loss and the light transport consistency loss, with weights $\lambda_{\text{diffusion}}$ and $\lambda_{\text{consistency}}$, respectively. During the inference stage, we can obtain the corresponding controllable image relighting results by inputting an image along with an appropriate background reference.

Specifically, IC-Light shows a very interesting application by manually defining a gradient mask as the potential light map, as the background reference. Then, it can be used to generate the light-controllable results in a specific direction. This light map reflects the positional and directional preference information of the illumination in the relighting results, laying the foundation and inspiring our controllable video relighting task.

## 3.2 LIGHTCTRL

Unlike image relighting methods that focus on single images and the specific lighting direction, our approach targets high-quality video relighting, particularly with a continuously varying light map, ensuring coherence and naturalness throughout the video. Specifically, given a source video, a relighting prompt, and a light map condition, our model aims to generate a new illumination for the video while preserving its original content. We begin by presenting the overall pipeline of the proposed framework, followed by a detailed description of its two key components: the Geometry-Aware Relighting module and the Light Map Injection module.

### 3.2.1 OVERALL PIPELINE

LightCtrl follows a similar approach to the training-free video editing paradigm (Meng et al., 2021), where the source video is first encoded into a noisy latent, and controllable relighting is achieved

through progressive denoising in the latent space. As shown in Fig. 2, we first encode the $l$ frame source video $\mathbf{V}_s$ into a latent space $\hat{\mathbf{z}}_0$, and add $T_m$ steps of noise to obtain the noisy latents $\hat{\mathbf{z}}_m$. Subsequently, according to the trajectory defined by the user, the corresponding frame-wise mask sequence is generated. For controllable relighting, firstly, we use the Light Map Injection module to obtain the trajectory-injected latents $\mathbf{z}_m$ from $\hat{\mathbf{z}}_m$, so that the following $T_m$ steps of denoising have the prior of generating the corresponding light trajectory. Then, similar to previous work (Zhou et al., 2025), we perform a single frame relighting and utilize the VDM for temporal consistency. In detail, during each denoising step $t$, we first predict the noise-free latent at 0 step $\hat{\mathbf{z}}_{0\leftarrow t}$ . Here, each frame of $\mathbf{z}_{0\leftarrow t}$ is decoded to the RGB space, which serves as the video consistent target $\mathbf{I}_t = \mathcal{D}(\mathbf{z}_{0\leftarrow t})$ using the pre-trained VAE decoder $\mathcal{D}(\cdot)$ for frame-wise relighting. To address detail loss, we calculate the initial detail residual $\Delta\mathbf{d}$ between the decoded latent in the first step $\mathbf{I}_m$ and the source video $\mathbf{V}_s$ pixelwisely and add it into $\mathbf{I}_t$ as the input of Geometry-Aware Relighting module for frame-wise relighting, obtaining the relighted target $\mathbf{I}'_t$ for the $t$-th denoising step. Finally, we employ a progressive fusion strategy with a fusion weight $\lambda_t$ that decreases as denoising progresses. The fusion target $\bar{\mathbf{I}}_t$ for the step $t$ is calculated as:

$$\bar{\mathbf{I}}_t = (1 - \lambda_t)\mathbf{I}_t + \lambda_t\mathbf{I}'_t. \tag{2}$$

The fusion latent $\bar{\mathbf{z}}_{0\leftarrow t} = \mathcal{E}(\bar{\mathbf{I}}_t)$ guides the next step of the denoising process towards the relighting direction. This ensures temporally coherent lighting, addressing temporal inconsistencies and enhancing the visual quality of the relighted video.

### 3.2.2 USER-DEFINED TRAJECTORY AS LIGHT MAP CONTROL SIGNAL

As mentioned above, the first key component of our framework enables the injection of a user-defined light trajectory, allowing the generated video to exhibit temporally varying lighting across frames that follows the specified trajectory. For example, with circular light sources, the user manually specifies a light trajectory along with the starting and ending radius. We then linearly interpolate the radius and position of the light for each frame to construct a frame-wise light map. This control signal is subsequently integrated into both the single-frame relighting process and the video diffusion denoising loop. Firstly, in the single frame relighting, as discussed in Sec. 3.1, IC-Light (Zhang et al., 2025) provides a manual background map to generate the relighting results in the corresponding direction. Inspired by this finding, we expand this directional background light source to the local dynamic for relighting video with differences as shown in the Fig. 3.

Although the above method can achieve some kind of controllability at the frame level, it still shows limited effects when facing the multi-step VDM denosing. Therefore, to produce more controllable results, inspired by the previous bounding box injection method (Qiu et al., 2024b) on leveraging the noise to guide the motion of objects in video generation, we adapt the concept to control the movement of illumination and then propose the Light Map Injection module.

During the video diffusion processes, given the initial noisy latent $\hat{\mathbf{z}}_m$ of the source video, we initialize a random Gaussian noise $\epsilon_{random}$ based on the area defined by the input light masks $\mathbf{M}$. However, directly replacing the noisy latents $\hat{\mathbf{z}}_m$ with the masked local latent $\epsilon_{random}$ caused severe visual artifacts. To mitigate this, we introduced a weighted fusion strategy. Mathematically, for each frame $k$, the new initial noise $\mathbf{z}_m^k$ is calculated as:

$$\mathbf{z}_m^k = \begin{cases} \hat{\mathbf{z}}_m^k & \text{if } \mathbf{M}_k[i,j] = 0 \\ \omega \cdot \epsilon_{\text{random}} + (1 - \omega) \cdot \hat{\mathbf{z}}_m^k & \text{if } \mathbf{M}_k[i,j] = 1, \end{cases} \tag{3}$$

where $\hat{\mathbf{z}}_m^k$ represents the noisy latent variable corresponding to frame $k$, and $\mathbf{M}_k$ denotes the input light mask at frame $k$. $\mathbf{M}_k[i,j] = 1$ when the position $[i,j]$ is inside the mask area of the light map, and $\mathbf{M}_k[i,j] = 0$ otherwise. $\omega$ is the fusion weight carefully tuned to balance the influence of the local random noise and the original noise, ensuring a smooth transition and minimizing artifacts. Based on the above strategy, the light trajectory can be successfully injected into both the video diffusion model and the image diffusion model for controllable video relighting.

### 3.2.3 GEOMETRY-AWARE RELIGHTING MODULE

Directly using the proposed Light Map Injection module can achieve controllable video relighting. However, the light environment in the source video often causes undesirable leakage into the final

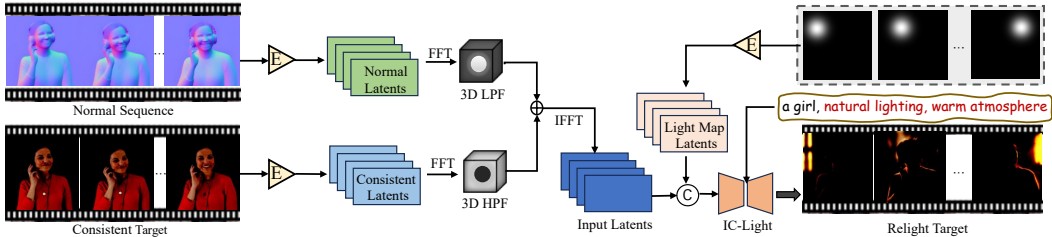

Figure 3: **The framework of Geometry-Aware Relighting Module**. This module is designed for frame-wise relighting. Firstly, the normal sequence of the source video and each consistent target derived from the denoising steps are encoded into the latent space and further process them in the frequency domain. Then, their spatio-temporal low-frequency components are dynamically fused with the high-frequency components of the consistent latents, with the cut-off frequency varying at each denoising step to add details. Finally, these lantents are transformed back to the video space and produce the relighted frames using frame-wise IC-Light Zhang et al. (2025).

output. For instance, when the illumination direction changes from left to right, artifacts may persist where the left side of the subject's face remains incorrectly lit. To address this, we propose incorporating geometry information(surface normal maps) as a part of single-frame relighting.

Similar to the RGB image, we find that the latent of the normal map contains stronger structural information and can also be utilized in a zero-shot setting. However, directly integrating this geometry information before each step of relighting would bring the blurring results caused by the pre-trained model, thereby severely affecting the quality and details of the final relighted video. Thus, we involve the frequency features for better fusion. As shown in Fig. 3, specifically, in each step of VDM denoising, we firstly use a pre-trained state-of-the-art normal estimation model, Stable Normal (Ye et al., 2024), to predict the frame-wise normal map $\mathbf{N}$ from the source video. Given the consistent target $\mathbf{I}_t$ as input to the relight model, we first encode it into the latent space to obtain the consistent latents $\mathbf{z}_{0 \leftarrow t}$. We perform the similar VAE encoding to the predicted normal $\mathbf{N}$, obtaining $\mathbf{z}_{normal}$. Then, we transfer them into the frequency domain via the Fast Fourier Transformation $\mathcal{FFT}_{\text{3D}}$ in the spatial and frequency domain (Wu et al., 2024). Whereas we define a dynamic 3D Butterworth filter $\mathcal{H}_\alpha(t)$ with cut-off frequency at $\alpha$ to control the high-frequency and low-frequency details between two latents. The detailed definition of the filter will be provided in Appendix B. After that, we combine the low-frequency latents and high-frequency latents and recover them into the spatial latent domain via inversed 3D Fast Fourier Transformation $\mathcal{IFFT}_{\text{3D}}$, which serves as the input latents $\tilde{\mathbf{z}}_t$ for frame-wise IC-Light relighting. Formally, this process can be written as:

$$\tilde{\mathbf{z}}_t = \mathcal{IFFT}_{\text{3D}} \left( \mathcal{FFT}_{\text{3D}}(\mathbf{z}_{normal}) \odot \mathcal{H}_\alpha(t) + \mathcal{FFT}_{\text{3D}}(\mathbf{z}_{0 \leftarrow t}) \odot (1 - \mathcal{H}_\alpha(t)) \right), \quad (4)$$

As the denoising loop progresses, we linearly decrease the cut-off frequency $\alpha$ so that the dynamic filter $\mathcal{H}_\alpha(t)$ gradually changes from all-pass to low-pass. In this way, we completely adopt the normal latents as input at the beginning of denoising. In the later stage of denoising, we utilize more of the low-frequency part of the normal latents and gradually add the high-frequency information from the consistent latents. This module enables us to effectively leverage the normal information to eliminate interfering lighting distributions while preserving details.

## 4 EXPERIMENTS

### 4.1 EXPERIMENTAL DETAILS

**Baselines.** As a baseline, we adopt IC-Light (Zhang et al., 2025), a state-of-the-art controllable image relighting model that relights each frame individually based on its corresponding light map from the input lighting trajectory. To balance frame-wise relighting controllability with temporal consistency, we also develop another baseline method based on SDEdit (Meng et al., 2021), where the frame-wise relighted video produced by IC-Light is perturbed with noise (using two levels: 0.2 and 0.6) and subsequently denoised. Finally, we compare against a variant based on the recent video relighting method Light-A-Video (Zhou et al., 2025), which we refer to as LAV-Traj, in which the light map from the input lighting trajectory is provided as the background image to the frame-wise image relighting model used in Light-A-Video.

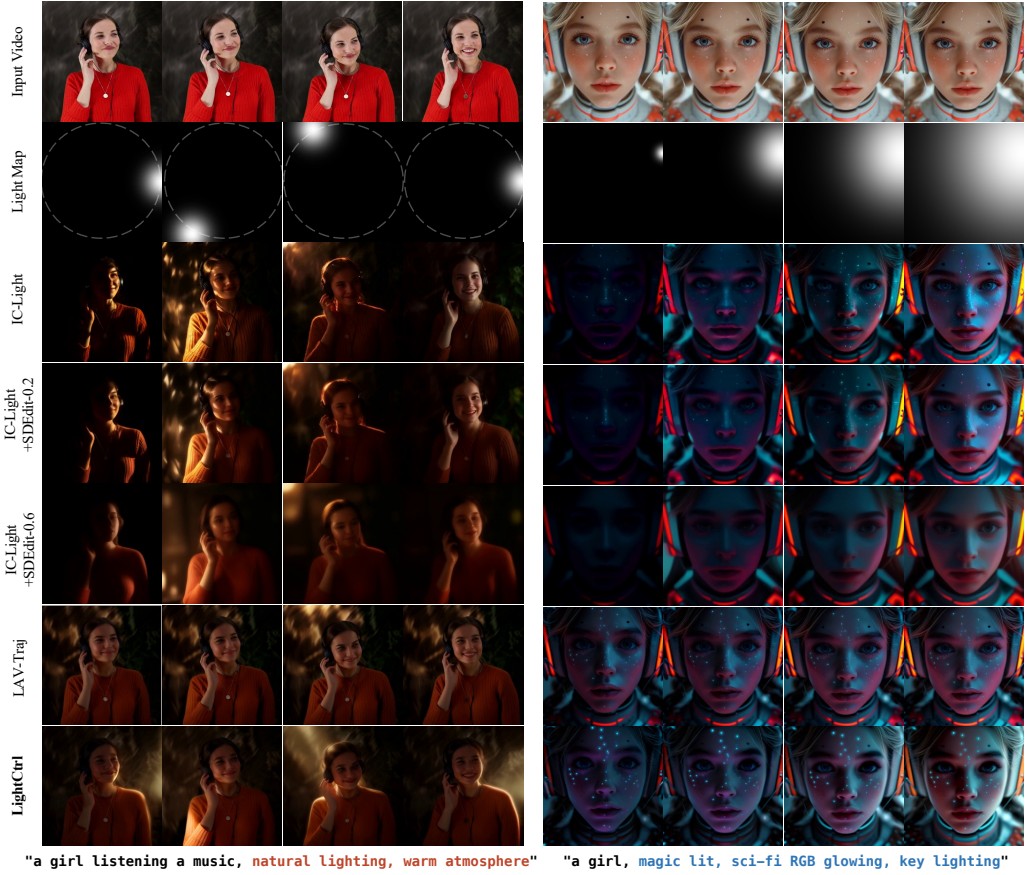

Figure 4: Qualitative comparison of light trajectory control. We compare our LightCtrl with IC-Light, IC-Light + SDEdit-0.2, IC-Light + SDEdit-0.6, and LAV-Traj. Compared to baselines, LightCtrl produces high-quality, temporally coherent, and controllable relighting results, closely following the input light trajectories.

**Datasets.** We construct a test dataset comprising 50 videos, primarily sourced from Pixabay (pixabay, 2025), which offers a diverse collection of high-quality footage. For each video, we apply six predefined light trajectories featuring variations in light source radius and trajectories, including left-to-right, top-to-bottom, circular motion, and more. Additional results and comparisons are provided in Appendix D.

**Evaluation Metrics.** We conduct quantitative evaluations to assess the relighting controllability with respect to the input light trajectory, as well as the visual quality. For the light trajectory controllability, we report the $\text{PSNR}_y$ in the $y$ channel(the luminance channel) in the light map mask region between the results of the baseline methods and a pure white video to measure the controllability of illumination. When the result in the mask area of the light map is closer to a pure white result in the luminance channel, we have reason to believe that there is more lighting in that area, which also reflects a better ability to control the video lighting according to the lighting trajectory. Then, we directly overlay the light map on the original video and calculate the $\text{PSNR}_{light}$ between the overlay results and all baseline methods to evaluate the controllable temporal consistency of lighting and foreground. Secondly, to evaluate the video quality of the results, we adopt aesthetic quality in Vbench (Huang et al., 2024) to assess the aesthetic quality of videos, and calculate the Frechet Video Distance (FVD) (Unterthiner et al., 2018) scores between all video results of different baselines and the input videos.

Finally, we conduct a user study with 40 volunteers. We randomly select 24 videos from the 50 videos generated by each method, where these 24 sets of videos include 6 predefined trajectories by users, as well as source videos with different characteristics and a variety of lighting prompts. They assessed the performance of our method by comparing it with baseline methods across four key aspects. Specifically, they examined **V**ideo **S**moothness (the coherence and naturalness of appearance

Table 1: Quantitative Comparisons with Other Methods.

| Methods | Video Quality | | Control Ability | | User Study | | | |
|---|---|---|---|---|---|---|---|---|
| | AQ ↑ | FVD ↓ | $\text{PSNR}_y$ ↑ | $\text{PSNR}_{light}$ ↑ | VS ↑ | LC ↑ | LQ ↑ | ALT ↑ |
| IC-Light Zhang et al. (2025) | 0.5937 | 1018.5 | 11.059 | 15.850 | 1.00% | 15.00% | 3.00% | 9.73% |
| IC-Light + SDEdit-0.2 Meng et al. (2021) | 0.5907 | 1134.8 | 11.009 | 16.249 | 2.50% | 2.00% | 0.50% | 3.00% |
| IC-Light + SDEdit-0.6 Meng et al. (2021) | 0.5681 | 1630.9 | 10.980 | 16.385 | 4.75% | 1.09% | 0.50% | 3.27% |
| LAV Zhou et al. (2025)-Traj | **0.6157** | 1077.4 | 11.043 | 17.755 | 23.50% | 4.00% | 20.00% | 10.27% |
| **LightCtrl (Ours)** | 0.6114 | **993.1** | **11.768** | **18.532** | **68.25%** | **77.91%** | **74.86%** | **73.73%** |

and lighting changes in the video), **L**ighting **C**ontrollability (the alignment of lighting movement with the given trajectory), **L**ighting **Q**uality (the overall quality and rationality of the lighting effects), and **A**lignment between **L**ighting and **T**ext (the match between the generated lighting and the provided relight prompt). The best results from the baseline methods for each dimension are chosen by the volunteers and used as preference metrics.

**Implementation Details.** In our experiments, following by (Zhou et al., 2025), we adopt IC-Light (Zhang et al., 2025) as the image relight model and employ AnimateDiff (Guo et al., 2023) as the default VDM. We typically set $T_m$ to 25. Then we set $\lambda_t = 1 - t/T_m$ in the progressive fusion strategy, where the fusion weight $\lambda_t$ decreases as denoising progresses. In the geometry-aware relighting module, we set the cut-off frequency $\alpha = \lambda_t$ so that as the denoising loop progresses, $\alpha$ is linearly decreasing. More implementation details are in Appendix A.

## 4.2 COMPARE WITH THE STATE-OF-THE-ART METHODS

Fig. 4 presents the control ability of illumination of all methods. The IC-Light per-frame model, while capable of producing high-quality relighting results on a frame-by-frame basis, performs poorly in terms of temporal consistency when dealing with continuously varying lighting trajectories. This results in significant flickering and discontinuity in the overall visual output. However, ICLight + SDEdit, although introducing motion priors through VDM, suffers from inconsistency in lighting after relighting at a noise level of 0.2, and excessive smoothing of appearance at a noise level of 0.6. Under the PLF fusion strategy, LAV-Traj has demonstrated strong capabilities in terms of temporal consistency in relighting results. However, when it comes to handling the diverse lighting trajectory variations we require, its controllability still falls short and fails to achieve controllable results for a variety of lighting trajectories.

The quantitative comparison of our method with other baseline methods is shown in Table 1. The IC-Light results still demonstrate their effectiveness in terms of the controllability of lighting in each frame. However, the obvious temporal inconsistency greatly reduces the aesthetic quality AQ score of the results. By combining SDEdit with different noise levels, temporal consistency is improved, but since it only smooths the video by changing the noise level and cannot achieve a consistency constraint on lighting, the video quality scores AQ and FVD, as well as the controllability metrics PSNR, are all low. LAV-Traj achieves the best aesthetic quality AQ score, but its strong consistency constraint greatly reduces its lighting controllability based on the given trajectory. In contrast, LightCtrl achieves a low FVD score and a high AQ score. While maintaining high temporal consistency in relighting quality, it also achieves the best scores in both controllability metrics $\text{PSNR}_y$, $\text{PSNR}_{light}$, demonstrating outstanding lighting control capability. Furthermore, we demonstrate our method's superiority under static light conditions in Appendix C.

## 4.3 ABLATION STUDY

We conduct an ablation study to evaluate the importance of different components. Considering the original LAV + Traj as baseline (Zhou et al., 2025), due to the strong temporal consistency constraint, it is unable to handle controllable continuous variations in lighting trajectories. As shown in Fig. 5, the Geometry-Aware Relighting module makes the lighting results on human faces more reasonable, independent of the lighting distribution in the source video. Meanwhile, the lighting distribution brought by the source video on the lawn on the right side causes the LAV-Traj to still highlight the background (as shown in the red box) when the light reaches the lower part of the frame, even when our method does not have the GAR module. In addition, by injecting lighting trajectories into

the initial noise before VDM denoising, LightCtrl can generate more controllable relighting videos while preserving the overall details of the video(as shown in the right part of Fig. 5). Due to the space limitation, we give more ablation experiments in Appendix B.

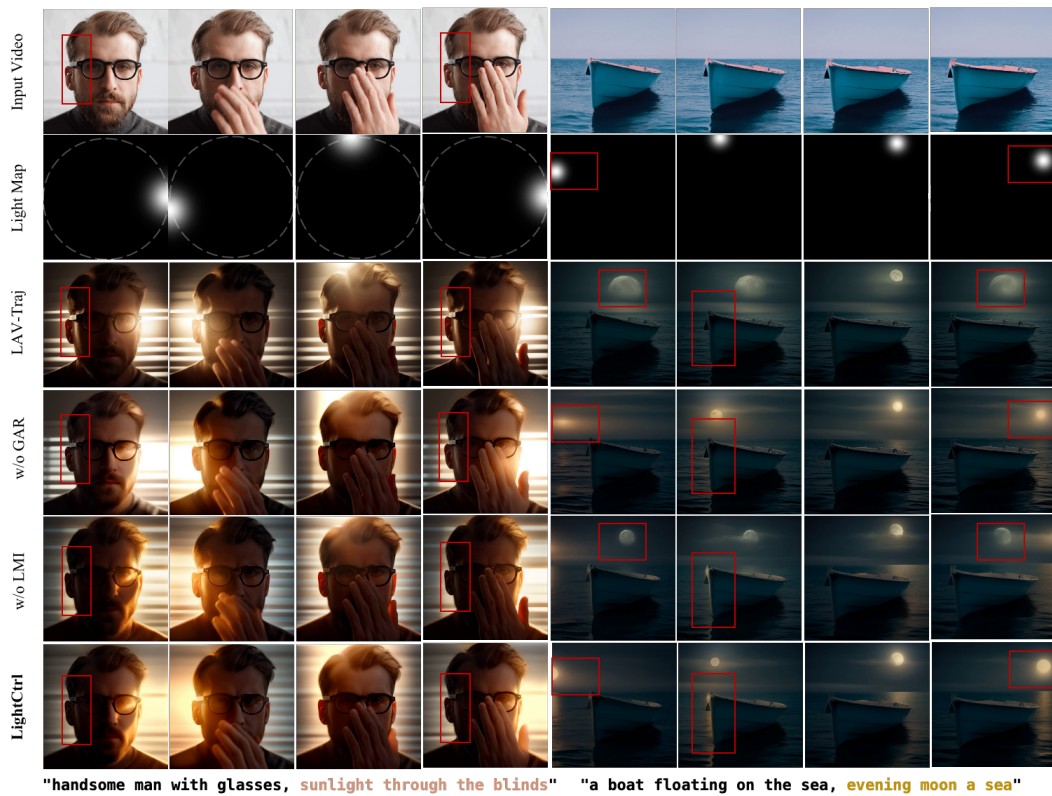

"handsome man with glasses, sunlight through the blinds"    "a boat floating on the sea, evening moon a sea"

Figure 5: **Ablation Study.** Results of controllable video relighting with the Geometry-Aware Relighting (GAR) module or the Light Map Injection (LMI) module removed. The red box indicates the light distribution brought by the input video.

## 4.4 LIMITATION AND FUTURE WORK

Since our method is based on existing image relighting models and VDM models, although our approach can achieve controllable lighting results for different light paths under the zero-shot setting, its performance is still limited. For example, when the light path passes over the foreground, it can cause abnormal flickering on the face. Moreover, our method currently cannot achieve 3D perception of the light path. Despite the use of geometry-aware information, we still cannot effectively remove the influence of obvious shadows in the source video. As shown in Fig. 6, the shadow on the right side of the cat remains after relighting. To address these limitations, we will adapt the latest video

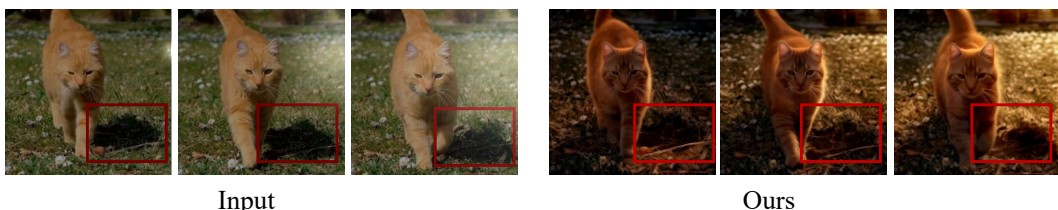

Input                                        Ours

Figure 6: **Failure Case.** We show an example result where the light map is blended into the original input video. Our method struggles to perform effectively in scenarios involving strong shadows, such as the "cat" case with the relight prompt "a cat, warm sunshine."

generation base models and design an advanced network in our future work to achieve higher-quality controllable lighting.

## 5 CONCLUSION

We introduce LightCtrl, a controllable video relighting method using pre-trained video and image diffusion models. Thus, the image relighting model produces high-quality relighting results, whereas the video diffusion model provides robust temporal consistency and control over the entire video via user-supplied trajectories. To achieve this goal, we propose two key modules, Light Map Injection and Geometry-Aware Relighting, to boost controllability by enhancing illumination coherence and reducing the impact of initial lighting. LightCtrl precisely generates high-quality videos with diverse illumination changes according to light trajectories, outperforming baseline methods in controllability as well as maintaining the temporal consistency.

**Acknowledgments**. This work was financially supported in part by National Natural Science Foundation of China (Project No. 62506064) and Guangdong Provincial Regional Joint Fund (Project No. 2024A1515110052).

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

## A  IMPLEMENTATION DETAILS

**Inference time.** We automatically generate light masks consistent with the video length based on user-defined trajectory patterns. On our A6000 GPU, the inference time for a 512×512 video with 16 frames is approximately 240 seconds.

**Details of user study.** We used a Tencent survey to create a questionnaire for all volunteers to evaluate the generated 24 sets of videos from four aspects.

i) **Video Smoothness:** Does the appearance and illumination changes in the video flow coherently, naturally, and smoothly, without noticeable sudden flickers or jumps?

ii) **Lighting Controllability:** Does the movement of illumination in the video align with the given illumination trajectory?

iii) **Lighting Quality:** How is the overall lighting quality of the foreground and the entire frame, and is the lighting reasonable?

iv) **Alignment between Lighting and Text:** Is there consistency between the generated illumination in the video and the given relight prompt?

Fig. 7 illustrates an example of the questionnaire utilized in our study. For a comprehensive assessment, we recruited 40 participants, all participants were students working in video generation or computer vision related fields. Before the study, all participants received a concise tutorial explaining the relighting task and the meaning of each evaluation metric, including lighting controllability, so that they shared a consistent and informed understanding of what they were rating. In the provided study example, we also explicitly state the definition of each metric. To enhance efficiency, users were guided to compare multiple methods within each evaluation dimension. Prior to comparing our method with the baseline approaches, we randomly shuffled the order of videos for each evaluation metric. Participants were then asked to select the more favorable result for each comparison. To ensure the rationality and objectivity of the lighting controllability evaluation, we include the current lighting map in each comparison video. This allows users to refer to the movement trajectories of the lighting, which are based on six predefined lighting patterns, each with distinct characteristics.

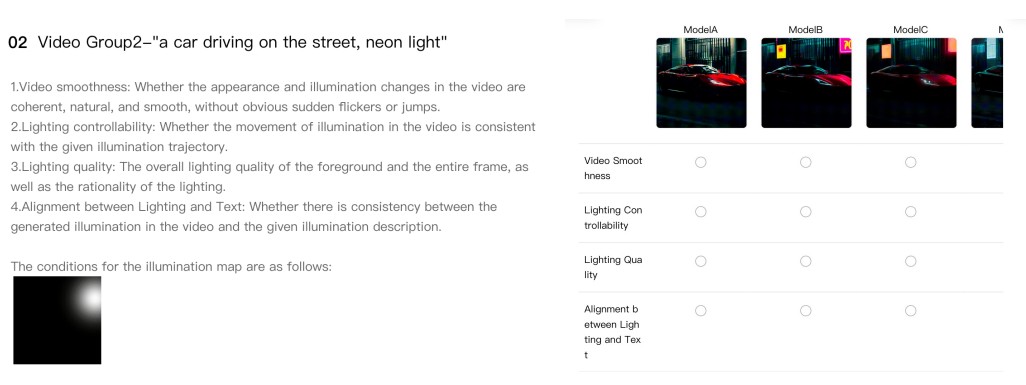

Figure 7: **Visualization of the questionnaire for user study.** Users are asked to compare the all baselines at each aspect and select their preferred one.

## B MORE ABLATION STUDIES

**The definition of the dynamic filter in the GAR module.** In the GAR module, to better fuse the normal map and the RGB latent, we adopt a strategy of dynamic fusion in the frequency domain. The definition of this dynamic filter $H_\alpha(t)$ is as follows:

$$H_\alpha(t) = \begin{cases} \frac{1}{1 + \left( \frac{D^2(t,h,w)}{d_s(t)^2} \right)^n} & \text{when } D(t,h,w) \leq d_s(t) \\ 0 & \text{when } D(t,h,w) > d_s(t) \end{cases} \tag{5}$$

where **the time-varying cut-off frequency:** $d_s(t) = d_t(t) = \alpha = 1 - \frac{t}{T_m}$, the frequency distance calculation is updated to:

$$D(t,h,w) = \sqrt{\left( \frac{d_s(t)}{d_t(t)} \cdot \left( \frac{2t}{T} - 1 \right) \right)^2 + \left( \frac{2h}{H} - 1 \right)^2 + \left( \frac{2w}{W} - 1 \right)^2} \tag{6}$$

Through the above method, we have defined a 3D dynamic filter, whose cut-off frequency is a value that changes with the denoising step $t$. By gradually adjusting the high-frequency and low-frequency information, we have effectively injected the normal map.

**Impact on feature selection of the GAR module.** In our GAR module, we introduce geometry feature information to mitigate the bias in lighting distribution brought by the source video. In our experiments, we adopt two types of features: delight images and normal maps. However, since we use pre-trained models for these features, the domain-specific models' performance significantly affects the quality, impacting the subsequent relighting results. As shown in Fig. 8, we use two delight models to remove the lighting distribution in the source video. While both the StableDelight and IntrinsicAnything models can mitigate the illumination distribution in the input video to some extent, noticeable artifacts (highlighted by the red boxes) emerge in the generated results after injecting information via the GAR module, as evident from the third and fourth rows. In contrast, our method, by incorporating normal information, produces more plausible and high-quality outputs. Moreover, as demonstrated in the third frame, our approach excels in terms of lighting quality in the generated results. Overall, our results can better reflect the impact of lighting changes in the video. Therefore, in our experiments, we chose the normal map with better feature quality.

**Impact on injection method of the GAR module.** After selecting the normal map as the input for the GAR module among delight images and the normal map, we need to consider how to integrate this feature into our model. Here, we conducted an ablation experiment on the injection method. We attempted direct weighted fusion, using half of the consistent target and half of the normal map as the input. However, due to the loss of details in the normal map, the result of direct fusion showed significant blurriness (the third left row of Fig. 9). Then, we fused the two in the frequency domain, combining the high-frequency information of the consistent target, which is rich in detail,

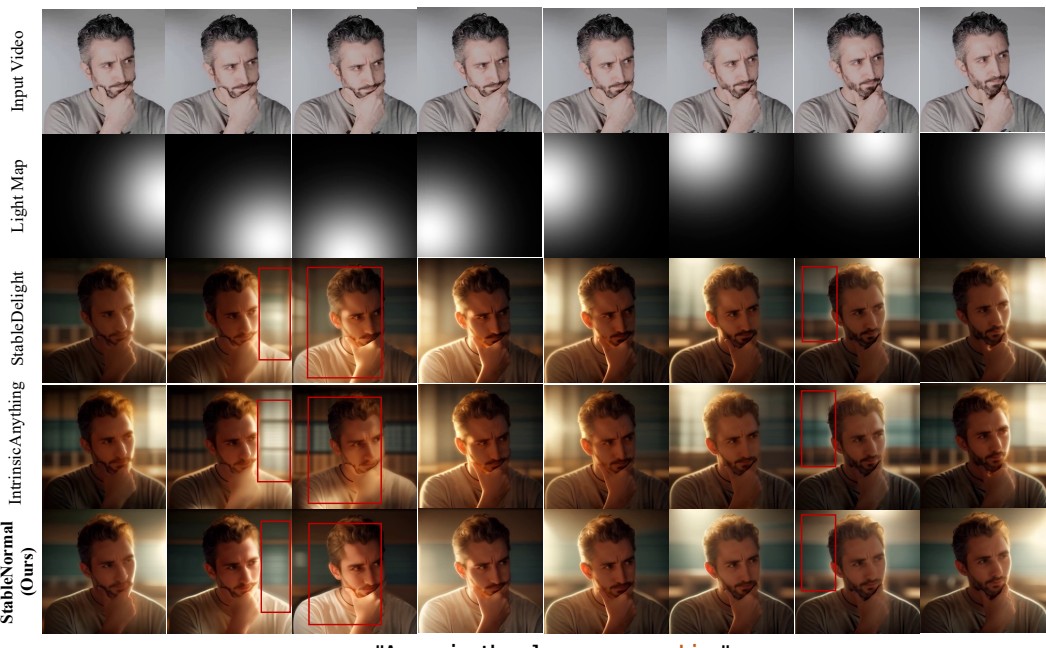

Figure 8: **Ablation Study on Feature Selection of GAR module.**

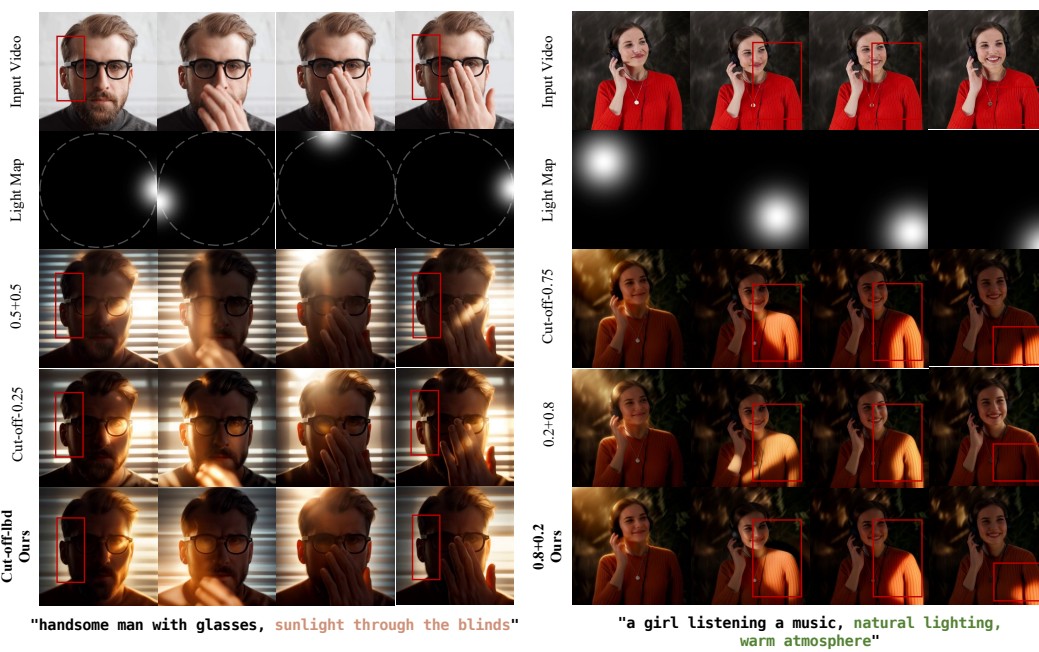

Figure 9: **Ablation Study on Injection Method of GAR(Left) and LMI module(Right).**

with the low-frequency information of the normal map. But from the results (the fourth and fifth left rows of Fig. 9), we found that no matter how we adjusted the cut-off frequency of the filter, we couldn't effectively remove the highlight in the red box on the left side of the image. Based on this, we decided to adopt a dynamic fusion method, injecting more normal map information in the early stage of denoising and supplementing details with the consistent target in the later stage. Finally, we achieved a good balance between detail preservation and prevention of light

Table 2: Comparison of Normal Injection Strategies on Video Quality

| Method | AQ ↑ | FVD ↓ |
|---|---|---|
| 0.5 + 0.5 | 0.5892 | 1010.9 |
| Cut-off-0.25 | 0.6097 | 998.7 |
| Cut-off-lbd | **0.6114** | **993.1** |

leakage(the last left row of Fig. 9). To further validate the benefit of this design, we additionally include another set of comparisons between latent space and frequency space integration as the right part of Fig. 11. We evaluate video quality under both strategies, and the results are consistent with the visual observations: the frequency-domain method yields markedly superior performance across all quantitative metrics, as reported in Table 2 .

**Impact on injection method of the LMI module.** When injecting the light map, to strike a balance between detail preservation and post-injection lighting controllability, we explored two fusion methods: direct fusion in the latent space and frequency-domain fusion of local noise and the original noisy latent, separating high and low frequencies. Our experiments revealed an issue with the frequency-domain approach. When using the original noisy latent to supplement high frequencies after injecting local noise to maintain details, the edges of the girl's sweater under sunlight exhibited excessive (as shown in the third row of the right image in Fig. 9). This over-enhancement of details deviated from our desired outcome. Consequently, we adopted a direct weighted fusion strategy in the latent space. We conducted multiple experiments to determine the optimal weights for local and original noise. Initially, when the original noise accounted for 0.8 (as shown in the fourth row of the right image in Fig. 9), while the excessive outlining was mitigated, lighting controllability decreased. For instance, in the last figure, although lighting was present in the bottom-right corner, the corresponding illumination effect was not properly reflected in the image. To address this, we increased the weight of local noise, ultimately settling on a ratio of 0.8 local noise to 0.2 original noise. This configuration effectively reduced excessive outlining while ensuring lighting controllability, making the lighting in the image more consistent with expectations(as shown in the last row of the right image in Fig. 9). It enables us to preserve image details while achieving precise adjustment and control of the lighting distribution.

Table 3: Quantitative Results between Raw Frames and Decoded Frames + Residual.

| Methods | AQ ↑ | FVD ↓ |
|---|---|---|
| Raw Frames | 0.5946 | 1037.4 |
| With Residual | **0.6114** | **993.1** |

**Raw Frames vs Decoded Frames + Residual.** This design compensates for the loss of local high-frequency information introduced by the VDM denoising process. The residual computed once at the first step restores these fine details while preserving the relighting effects progressively established through latent space denoising. Using the raw frames directly would break this balance. As demonstrated in Fig. 10, directly injecting raw frames causes the output video to inherit substantial illumination patterns and low-frequency structures from the input, which preserves appearance but ultimately fails to achieve correct relighting. In contrast, the residual formulation enables us to replenish structural details during the early denoising stages, while its influence gradually diminishes in later stages where lighting appearance is formed. This yields the desired balance: retaining necessary fine-grained details without interfering with the generation of high-quality, trajectory-controlled relighting. We also compute AQ and FVD for both, as shown in Table 3, further quantifying the improvement in video quality achieved by our design.

**The Effects of Progressive fusion.** The progressive fusion mechanism is key to achieving temporal consistency. As illustrated in the left part of Fig. 11, compared with the per-frame results produced by IC-Light, progressive fusion ensures that both the foreground and background remain noticeably coherent over time. In particular, the red-boxed background region shows that without progressive fusion, the background exhibits clear frame-to-frame fluctuations with no temporal stability, whereas our fusion strategy effectively suppresses such inconsistencies and maintains stable appear-

Table 4: Effect of Progressive Fusion on Temporal CLIP-Score

| Methods | CLIP-Score ↑ |
|---|---|
| w/o Progressive Fusion | 0.9200 |
| With Progressive Fusion | **0.9634** |

ance throughout the sequence. To objectively evaluate the improvement in temporal consistency brought by progressive fusion, we additionally report the average CLIP score across consecutive video frames, which reflects semantic and structural coherence over time. In the Table 5 with progressive fusion, the model achieves higher CLIP scores, confirming its contribution to smoother and more stable video relighting.

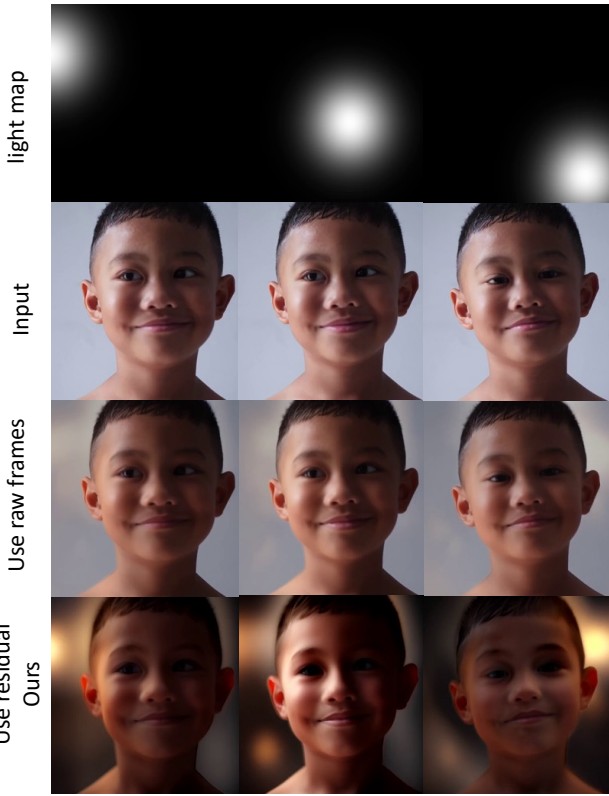

Figure 10: **Ablation Study on Raw Frames vs Residual.**

## C MORE QUANTITATIVE RESULTS

**The effectiveness of static light conditions.** We have also conducted a experiment evaluating illumination control in static scenes (Specifically, we keep the light map in the same position throughout the entire video) using our established $PSNR_{\text{light}}$ metric (defined in Section 4.1 and used for temporal consistency). The results show our method achieves **17.494 dB**, **exceeding all baselines** (next-best Zhou et al. (2025) 16.630 dB). This quantitative evidence confirms our method's lighting control superiority even in static conditions, supporting our original claim. The complete results are presented as shown in Table 5:

## D MORE QUALITATIVE RESULTS

**More results based on CogVideoX.** In Fig. 12, we present some results on the DIT-based video diffusion model, such as CogVideoX. These results also demonstrate that our model can be adapted

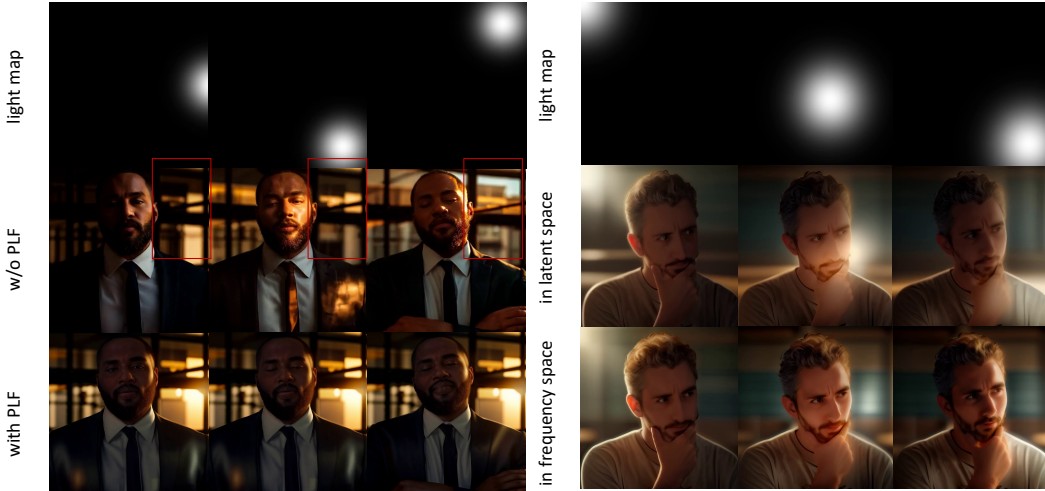

Figure 11: **Ablation Study on Progressive Fusion and Injection method of Normal.**

Table 5: Comparison of $PSNR_{\text{light}}$ (dB) under static light conditions

| Methods | IC-Light | SDEdit-0.2 | SDEdit-0.6 | LAV-Traj | Ours |
|---|---|---|---|---|---|
| $PSNR_{\text{light}}$ | 12.112 | 13.349 | 13.686 | 16.630 | **17.494** |

to multiple video diffusion models. Since the model of CogVideoX uses the DDIMScheduler, while our main model, based on AnimateDiff, uses the EulerAncestralDiscreteScheduler, the intensity of the lighting colors in the final results is largely affected by the different schedulers. The lighting results from the DDIMScheduler tend to be lighter. This difference in color intensity is also a reasonable phenomenon.

**More results of LightCtrl.** In Fig. 14, we present more visual results of LightCtrl. These examples show that our model can generate controllable video relighting under different customized trajectories in various scenes.

## E    LARGE LANGUAGE MODEL USAGE

In the preparation of this work, the authors used LLMs (e.g., GPT-4, Claude, etc.) solely for text polishing, including grammar checking and sentence refinement. The LLM did not contribute to the scientific content, and the authors assume full responsibility for the work.

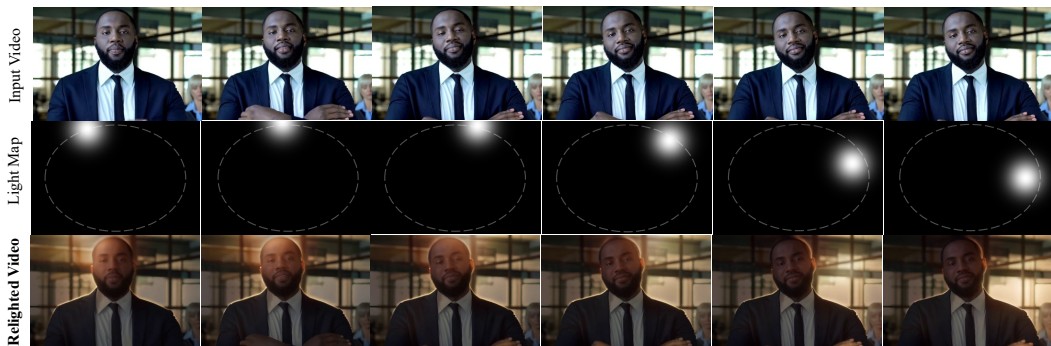

*Vdm prompt:* *"a man is standing in front of a window"*

*Relight prompt:* *"a man is standing in front of a window, warm sunlight through the window"*

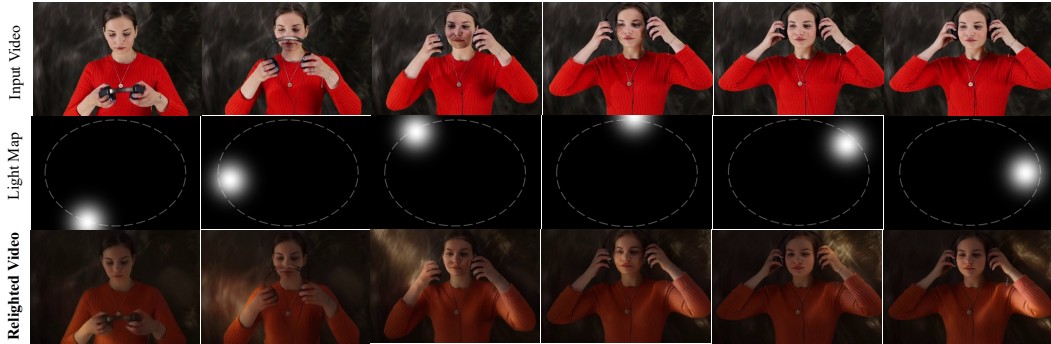

*Vdm prompt:* *"a girl listening a music"*

*Relight prompt:* *"a girl listening a music, natural lighting, warm atmosphere"*

Figure 12: **More results based on CogVideoX.**

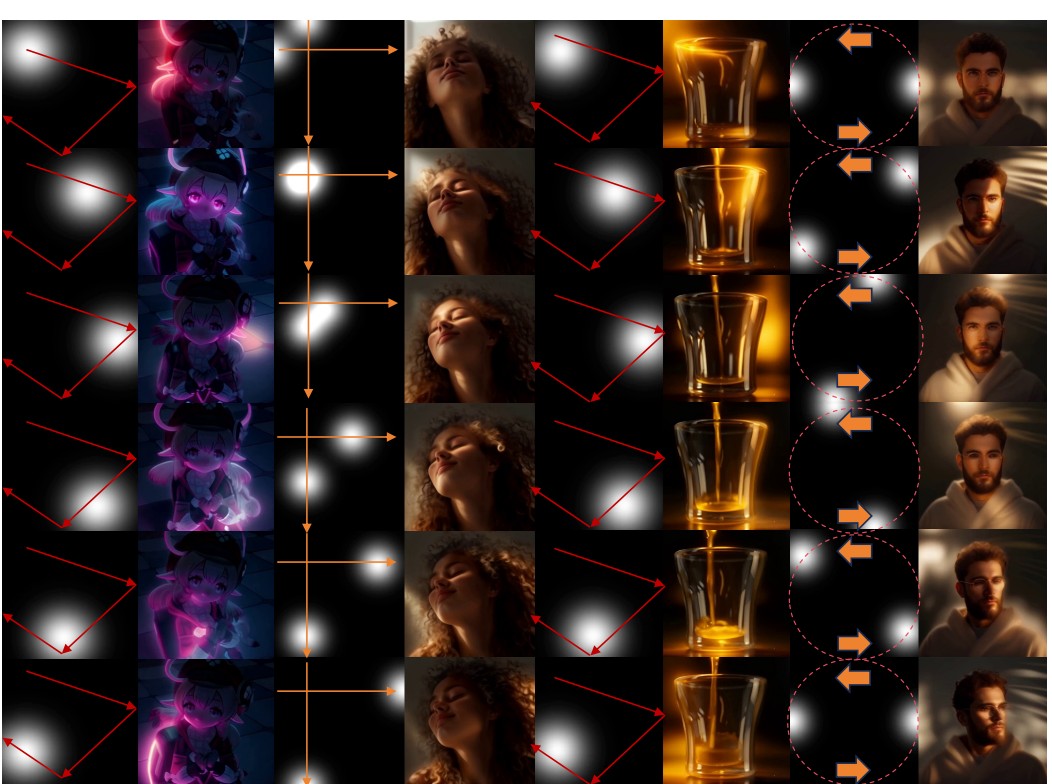

Figure 13: **More Results with Diverse Lighting Trajectories.** The first and third columns show the results under random polyline illumination trajectories. The second column presents the results of two light sources moving independently, one from left to right and the other from top to bottom. The fourth column shows the results of two light sources moving in opposite directions along circular paths.

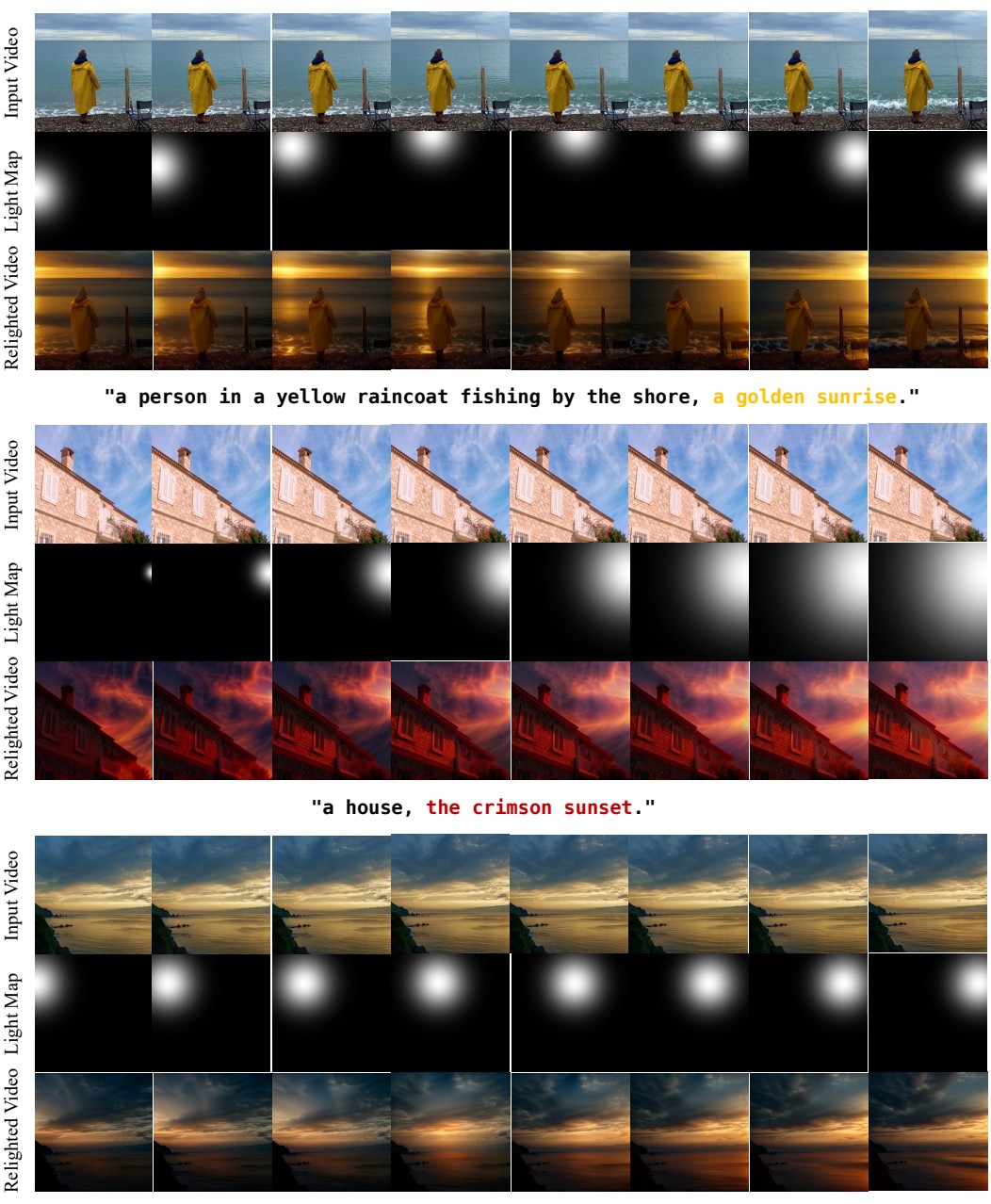

Figure 14: **More results of LightCtrl in light trajectory-conditioned video relighting .**

