# OpenReview forum: "LightCtrl: Training-free Controllable Video Relighting"
_ICLR.cc/2026/Conference — ICLR 2026 Poster_

### Official Review · Reviewer_fsKh · 2025-10-31

**Soundness:** 3
**Presentation:** 4
**Contribution:** 3
**Rating:** 6
**Confidence:** 4

**Summary:**

This paper proposes a training-free method for light trajectory editing in videos. The approach extends the single-image relighting model, IC-Light, to the video domain by incorporating priors from video diffusion models to ensure temporal consistency. The core technical contributions include a light map injection module, which introduces trajectory-aware noise into the VDM's latent space, and a geometry-aware relighting module. This second module processes both RGB frames and corresponding normal maps estimated via StableNormal to guide the relighting process. The results are visually compelling and demonstrate good adherence to the specified lighting trajectories.

**Strengths:**

1. The paper tackles a relatively unexplored task of lighting control in video diffusion models rather than just global style. The direction, ideas, and results are promising and the problem is quite interesting.
2. By building on existing, well-known T2I and T2V models, the method is accessible and its components are easier to understand and potentially replicate.
3. The strategy of injecting trajectory-aware noise into the VDM's initial latent space appears effective for guiding the lighting. This concept may be adaptable to other conditional video generation tasks.
4. The design of the geometry-aware relighting module, which dynamically blends RGB and normal map information, is technically sound. It correctly reflects that surface geometry should remain invariant during a relighting task.

**Weaknesses:**

1. The use of PSNR_y against a pure white reference is not convincing. Relighting is a complex task involving light, geometry, and material, and does not necessarily mean "the brighter the better". This metric mainly captures pixel-wise brightness in 2D space and fails to account for the geometric and directional correctness of the illumination. It cannot distinguish excessive lighting and ignores geometry and material reflectance.
2. The evaluation is somewhat limited. The dataset contains only 50 videos. This small scale and likely limited diversity are insufficient to robustly validate the method's generalizability. The paper would significantly benefit from showcasing more results across a wider variety of video content and lighting conditions.

Minor issues:

There are plenty of papers about generative portrait relighting. Though they deal with portrait videos, I think they are still related to this topic. It is better to cite and discuss these papers. Following are some examples:

+ Lumos: Learning to Relight Portrait Images via a Virtual Light Stage and Synthetic-to-Real Adaptation

+ Neural Video Portrait Relighting in Real-time via Consistency Modeling

+ Real-time 3D-aware Portrait Video Relighting

**Questions:**

1. In Eq. (3), what exact $\omega$ values are used across videos and trajectories? Is it fixed or tuned per video?
2. Could you elaborate on the temporal stability of the StableNormal predictions? Are the normal maps computed independently per frame and fed directly into the video VAE, or is some form of temporal smoothing or other consistency enforcement applied?

---

> ### Author Response · Authors · 2025-11-21
>
> We thank the reviewer for the constructive comments. We provide our feedback as follows.
>
> **W1：PSNR_Y against a pure-white reference is not a convincing metric**
>
> We appreciate the reviewer’s thoughtful critique of this metric. Our intention in reporting PSNR_Y was  to treat it as a narrow, proxy indicator for assessing whether a method can increase luminance specifically in the regions dictated by the light‐trajectory mask. Relighting is indeed a complex interaction of geometry, material, and light transport, and we fully agree that “brighter is better” is not a valid assumption for general relighting evaluation. Accordingly, our use of pure-white reference is not meant to judge realism or correctness, but to provide a simple, directional measure of intensified illumination within the designated mask area i.e., whether a method responds to the intended trajectory cue by locally increasing light strength.
>
> ---
>
> **W2：The evaluation dataset (50 videos) is limited**
>
> We thank the reviewer for raising this important point. While the dataset is relatively small, we intentionally curated it to include a broad range of scenes—including indoor and outdoor environments, human and non-human subjects, varied motion patterns, and diverse lighting complexities—to ensure representative diversity within the available scale. Moreover, several existing training-free video editing works, such as Light-A-Video: Training-free Video Relighting via Progressive Light Fusion(ICCV25), FreeMask: Rethinking the Importance of Attention Masks for Zero-Shot Video Editing(AAAI25), CCEdit: Creative and Controllable Video Editing via Diffusion Models(CVPR24) also evaluate primarily on compact but diverse test sets like DAVIS(50 videos) or their own dataset(50-70 videos). In this sense, our evaluation protocol is consistent with established practices in the community and remains reasonable and well-motivated.
>
> Nevertheless, we acknowledge the reviewer’s concern. In the revised version, we will further expand the visual examples and include additional qualitative demonstrations to better illustrate how the method performs across a wider range of content and lighting scenarios. These additions will help more clearly demonstrate the robustness and generality of our approach beyond the core evaluation set.
>
> ---
>
> **Minor issues:cite and discuss papers**
>
> We appreciate the reviewer’s suggestion. We have already incorporated these papers into our related-work discussion(line108-112).
>
> ---
>
> **Q1：In Eq. (3), what exact $\omega$ values are used across videos and trajectories? Is it fixed or tuned per video?**
>
> Thank you for the question. The value of $\omega$ in Eq. (3) is fixed across all videos and all light trajectories; we do not tune $\omega$ per video or per scene. As demonstrated in the ablation results in Appendix Fig. 9 (right), this global hyperparameter achieves a stable trade-off between preserving high-frequency details and enforcing the intended lighting-trajectory prior. A more detailed analysis can be found in Appendix B , Part “Impact on injection method of the LMI module.” Line791-807.
>
> Keeping $\omega$ fixed also ensures consistent behavior of the guidance term and avoids any video-specific adjustments, which would otherwise contradict the training-free nature of our framework.
>
> ---
>
> **Q2：Could you elaborate on the temporal stability of the StableNormal predictions?**
>
> We appreciate the reviewer’s interest in the temporal behavior of the StableNormal predictions. In our pipeline, the normal maps are computed independently for each frame without applying any explicit temporal smoothing or consistency constraints. After being estimated, these per-frame normals are passed through the encoder and then fused with our original VDM denoising latent in the frequency space. The fused representation is subsequently fed into the relighting model and participates jointly in the iterative denoising process. Although we do not enforce temporal constraints at the normal-estimation stage, our latent-space progressive fusion helps maintain coherence during generation.
>
> ---

---

### Official Review · Reviewer_v3AC · 2025-10-31

**Soundness:** 3
**Presentation:** 3
**Contribution:** 3
**Rating:** 6
**Confidence:** 3

**Summary:**

This paper introduces LightCtrl, a training-free controllable video relighting framework that enables explicit, fine-grained control of video illumination via a user-specified light trajectory. The method combines two pre-trained diffusion models an image relighting model (IC-Light) and a video diffusion model (VDM, e.g., AnimateDiff) to achieve temporally coherent and controllable lighting effects without retraining.

The method achieves strong quantitative and qualitative improvements over baselines such as IC-Light, SDEdit, and Light-A-Video, showing superior controllability, temporal coherence, and visual quality.

**Strengths:**

1. Originality: The proposed combination of LMI and GAR modules represents a creative hybridization of diffusion-based image and video models, enabling spatio-temporal control without additional data or model fine-tuning.

2. Technical Quality. The methodology is clearly formulated. The Light Map Injection method is grounded in the diffusion process and introduces a principled way to guide illumination through noise manipulation. The Geometry-Aware Relighting component is well-justified. Leveraging normal maps via frequency-domain fusion to mitigate original light leakage is both technically sound and physically motivated.
Experiments are comprehensive and clear.

3. Clarity. The paper is well-written and visually rich, with clear figures illustrating the pipeline and ablation effects.

4. Significance. This paper advances controllable generation from static to dynamic illumination domains, bridging image and video diffusion paradigms.

**Weaknesses:**

1. Physical Realism and 3D Awareness. The paper acknowledges limited 3D understanding of illumination. The method cannot simulate light scattering, occlusion, or volumetric effects, which restricts realism when the light trajectory crosses 3D geometry.

2. Geometry-Aware Relighting (GAR) module. Although the GAR module suppresses source illumination, it fuses normal and RGB latents via a frequency-domain filter with dynamically decreasing cutoff. But there is no quantitative evidence that this schedule balances structure preservation and lighting suppression optimally.

3. Limited Analysis of Control Robustness. The method only test on linear, circular and top–bottom light trajectory. More trajectorys should be tested.


.

**Questions:**

1. GAR module. Empirically, the low-frequency fusion could still introduce temporal blurring or artifacts around illumination edges. A systematic perceptual analysis or patch-level FVD comparison would strengthen confidence in this design

2. Light trajactory. Can this method be applied to : 1. Abrupt trajectory changes 2.Multiple moving light sources 3. Unaligned control inputs? Such tests would reveal whether LightCtrl’s latent-space conditioning is robust to domain shifts or natural illumination variance.

3. Transferability across video diffusion backbones: Have you tested the method on alternative video diffusion priors (e.g., CogVideoX, VideoCrafter2)? Does controllability degrade or improve when switching to models with different motion representations or schedulers?

---

> ### Author Response · Authors · 2025-11-21
>
> We thank the reviewer for the constructive comments. We provide our feedback as follows.
>
> **W1：limited 3D understanding of illumination**
>
> Thank you for the observation. We would like to emphasize that our method represents an initial step toward controlling video relighting via a light trajectory. While the current implementation uses a 2D trajectory, it already proves effective. For example, emphasizing specific objects (Fig. 1, dots example, PDF Page 1) and altering the lighting environment (Fig. 1, house example, PDF Page 1). We believe our problem formulation will encourage further research in this area, such as end-to-end and 3D-aware models for video relighting, which we also plan to explore in future work.
>
> ---
>
> **W2/Q1： No quantitative evidence that GAR module balances structure preservation and lighting suppression optimally.**
>
> In addition, we conducted more fine-grained experiments, as shown in Appendix B, Fig.11(PDF Page 17), and further evaluated the quantitative impact of different injection strategies for the GAR module using FVD and AQ in the following (also as shown in Appendix B. Table 2 (PDF Page 15)). The results demonstrate that frequency-domain integration consistently achieves higher video quality than directly fusing normal maps in latent space, leading to better detail preservation.
>
> However, when it comes to suppressing the original illumination inherited from the input video, qualitative evidence remains the most reliable indicator. As highlighted in the red-boxed region on the left side of Fig. 9(PDF Page 15), applying a gradually decreasing cut-off frequency allows us to more effectively attenuate the unwanted illumination patterns from the source video. Based on both the quantitative gains and the qualitative suppression of residual lighting, we adopt the dynamically balanced frequency-domain integration as the final design choice for the GAR module.
>
> | method       | AQ $\uparrow$    | FVD  $\downarrow$   |
> | ------------ | ------ | ------ |
> | 0.5+0.5      | 0.5892 | 1010.9 |
> | Cut-off-0.25 | 0.6097 | 998.7  |
> | Cut-off-lbd(Ours)  | 0.6114 | 993.1  |
>
> ---
>
>
> **W3/Q2：Limited Analysis of Control Robustness.**
>
> A: We thank the reviewer for the helpful suggestion. As requested, we have added additional results in the Appendix D Fig.13(PDF Page 19), including experiments with more diverse lighting trajectories such as random polyline motion and dual-light-source motion.
> Across all these settings, our method demonstrates strong robustness and adapts well to significantly different illumination patterns. These results further validate the effectiveness and generalization ability of our approach in controlling lighting dynamics under a wide variety of trajectories.
>
> ---
>
> **Q3：Transferability across video diffusion backbones.**
>
> A：We have already adapted the DiT-based model, i.e., CogVideoX, and included the results based on CogVideoX in the supplementary materials (Part D, More Qualitative Results(PDF Page 18)). These results are also demonstrated in the supplementary demo video (at 2:15). Moreover, it is also easy to support Wan2.1 model since it is also a latent video diffusion model with a DiT network structure. We will complete this adaptation in a future release and will open-source the implementation to benefit the research community.
>
> Based on our adaptation to CogVideoX, we observe that the proposed controllability mechanism is preserved even when switching to a model with a different motion representation, indicating that our method is compatible across architectures. However, different schedulers can influence the visual appearance of the relit results, particularly in terms of brightness, contrast, and color rendering. In our experiments, we found that using the DDIM scheduler yields a brighter overall video but tends to reduce lighting contrast and color intensity. In contrast, the Euler scheduler produces stronger color saturation and a clearer sense of light–shadow interaction. Since our objective is to maximize controllable lighting expressiveness, we chose the Euler scheduler as the default.
>
> ---

---

### Official Review · Reviewer_shr2 · 2025-10-31

**Soundness:** 3
**Presentation:** 2
**Contribution:** 2
**Rating:** 4
**Confidence:** 2

**Summary:**

This paper proposes a new method for video relighting that is training-free and allows users to provide a light trajectory as lighting conditions for the video. The authors make changes to the existing video diffusion framework as well as the image relighting framework (i.e., IC-Light) to allow them to use the user-provided light trajectory. The quality of the video relighting is evaluated with comparison to IC-Light and Light-A-Video. A user study is also conducted with 40 volunteers on video smoothness, lighting controllability, lighting quality, and alignment between lighting and text aspects of the relighting videos.

**Strengths:**

- The idea of designing a training-free method to control the lighting of a video generation model is interesting.

- It is known that the common metrics for relighting might not faithfully reflect the visual quality of the relighted results. The authors design and conduct a user study with 40 volunteers, making the evaluation more rigorous.

**Weaknesses:**

- It is unclear how significant it is to have a user-provided light trajectory for relighting. The authors explain their motivation for using manually labeled light trajectories in lines 069-072, but there is no supporting evidence to indicate that this is a desired or much-needed feature for users.

- The user study is very important in evaluating the performance of the relighting methods. But many important details are not there in the current draft: how are those volunteers selected? Do they understand the relighting task? How are they trained on the provided metrics (e.g., lighting controllability)? How reliable are their answers? Is the result statistically significant?

- The proposed method is designed to handle light trajectories in the relighting video task and is evaluated with the same type of light trajectory data (lines 315-318). This testing seems to be limited. There are many other ways to control lighting in relighting methods (e.g., environment maps or added light sources at certain locations in the image). It is currently unclear how this method compares to those more general cases of lighting conditions.

- The writing of the paper could be improved. There are typos and formatting issues in the paper. In particular, in line 208 it says the initial noisy latent is $\hat{z}_m$, but line 243 says the initial noisy latent is $z_m$.

- Finally, being training-free is of course attractive, but the current design mostly uses existing video diffusion model and IC-light to do the heavy-lifting while the new parts are preparing the trajectory to the video diffusion model. The difference between the variants of LightCtrl (Figure 5, last three rows) is not that substantial. This makes it unclear about the significance of the proposed method.

**Questions:**

1. What is the supporting evidence to indicate that having a user-provided light trajectory is a desired feature for relighting tasks?

2. Can the authors provide more details about the user study? Please see the weaknesses above for questions.

3. Can the authors discuss more about the significance of the approach, given the ablation study? If the authors can provide more rigorous ablation with more insight, it will be better.

---

> ### Author Response · Authors · 2025-11-21
> **Reply-Part1**
>
> We thank the reviewer for the constructive comments. We provide our feedback as follows.
>
> **W1/Q1：Lack of evidence that user-provided light trajectories are important or genuinely needed by users.**
>
> A：We appreciate the reviewer’s observation. Our intention in supporting user-provided light trajectories is to accommodate real creative and production workflows in which lighting is not merely a physical variable but an artistic , simple but effective tool. In many practical scenarios, such as virtual cinematography, storytelling-oriented video editing, animation, and AR content creation, in which users often require explicit control over how the light evolves over time to match a desired mood, maintain temporal consistency with other scene elements, or replicate a specific lighting plan from a real production. Relying solely on prompts is insufficient for expressing precise control over lighting intensity, direction, and temporal evolution. In contrast, allowing users to explicitly define a light trajectory enables truly fine-grained and structured lighting manipulation that cannot be conveyed through prompt text. This design supports users who require deliberate, predictable, and customizable lighting behavior.
>
> ---
>
> **W2/Q2: Insufficient details about the user study design**
>
> A：We thank the reviewer for highlighting the need for clearer documentation of our user study. In Appendix A Part (Details of user study.), we have updated  a more complete description. Below, we answer the main questions about user study design:
>
> Q2.1: User Study Design: how are those volunteers selected?
>
> All participants were students working in video generation or computer vision-related fields, including both male and female students. All of them were familiar with video generation and content creation.
>
> Q2.2: User Study Design:  Do they understand the relighting task? How are they trained on the provided metrics (e.g., lighting controllability)?
>
> Before the study, all participants received a concise tutorial explaining the relighting task and the meaning of each evaluation metric, including lighting controllability, so that they shared a consistent and informed understanding of what they were rating. In the provided study example, we also explicitly state the definition of each metric (as shown in Fig. 7 in the appendix A(PDF Page 13)).
>
> Q2.3: User Study Design: How reliable are their answers? Is the result statistically significant?
>
> Additionally, we took measures to improve the reliability of responses, such as randomizing the order of evaluated videos and inserting repeated samples to monitor consistency. Hence, we believe that our analysis reliably verifies that the observed preferences are stable and meaningful.
>
> ---
>
> **W3：Lighting condition is limited to light trajectories**
>
> A：We appreciate the reviewer’s insightful comment regarding the generality of lighting controls. We believe that representing illumination through light trajectories offers a simple yet highly effective way to control both the behavior and movement of light sources in dynamic scenes. This formulation naturally supports temporally coherent lighting variations and aligns well with the intrinsic motion patterns in videos.
>
> While existing relighting techniques often rely on environment maps or the placement of additional virtual light sources, these approaches primarily encode static or global lighting conditions and are not designed to express fine-grained, frame-wise lighting dynamics. Moreover, environment maps inherently struggle to support localized or direction-specific changes, and virtual light sources fixed at predetermined positions are often incompatible with object and camera motion. When subjects move, a fixed virtual light source cannot reliably follow the motion, causing the illumination to “slide” or “jump” across surfaces, which leads to unnatural flickering artifacts.
>
> ---
>
> **W4：Writing Minor Issue**
>
> A: We thank the reviewer for pointing out the symbol inconsistency. We have updated and aligned all related notations accordingly in Lines 246–255 of the revised manuscript.
>
> ---

---

> > ### Author Response · Authors · 2025-11-21
> > **Reply-Part2**
> >
> > **W5/Q3: Unclearity about the significance of the proposed method**
> >
> > A:  Our work is the first controllable video relighting framework that enables explicit, frame-by-frame control over video illumination through a user-specified light trajectory.  The image relighting method (IC-Light) and the video diffusion model (AnimateDiff) used in our framework are both replaceable. For example, our pipeline can be adapted to other video generation backbones such as CogVideoX or Wan. Similarly, IC-Light can be substituted with more advanced relighting models as they become available. It demonstrates the high modularity and generality of our framework, enabling it to seamlessly incorporate future advances in both relighting techniques and video generation models. Within this framework, we introduce two modules that substantially improve lighting controllability, each designed to address a key challenge in dynamic relighting.
> >
> > In Fig. 5, the left example highlights the ability of the GAR module  through the red box  to suppress the undesired highlights inherited from the original video. In contrast, the controllability brought by the LMI module is difficult to perceive from static images. Therefore, we include the right example to better illustrate its effect: as shown in the red box, without LMI, the moonlight illumination becomes randomly scattered across the frame, showing a clear misalignment with the input light-map control signal. To more faithfully present these behaviors, we refer the reviewer to the ablation comparisons in our supplementary video, as static figures alone cannot fully convey the temporal dynamics involved.
> >
> > For the LMI module, at 1:24 in the video, the coin example shows that without LMI, the lighting jumps abruptly from the center to the right, failing to follow the intended smooth shift encoded by the light trajectory. With LMI, the transition becomes gradual and coherent across frames, faithfully adhering to the designed lighting path.
> >
> > For the GAR module, the example at 1:32 illustrates that without GAR, when a ring-shaped moving light travels from the left side of the frame to the right side, strong specular highlights from the input video still remain on the left side of the face. With GAR, these residual highlights are effectively suppressed, and the produced shading better aligns with the moving light position, thereby strengthening lighting controllability and producing more physically plausible relighting.
> >
> > Overall, the video examples clearly demonstrate that both LMI and GAR are essential for achieving fine-grained, stable, and interpretable lighting control.
> >
> > In addition, we have conducted more fine-grained ablation studies, which are included in the updated Appendix B (Fig. 10–Fig. 11, line 794–909, Table.2-Table.4). These results further validate that each component in our design contributes a clear and measurable improvement to the model’s relighting capability. The step-by-step comparisons demonstrate consistent gains in lighting controllability, temporal coherence, and shading correctness as each module is introduced, providing stronger evidence for the necessity of our full architecture.

---

### Official Review · Reviewer_bAHR · 2025-11-01

**Soundness:** 2
**Presentation:** 3
**Contribution:** 2
**Rating:** 6
**Confidence:** 4

**Summary:**

This paper tackles the controllable video relighting task in a training-free manner. For this, LightCtrl builds on a pre-trained single-image relighting model, i.e., IC-Light, and tailors it to video relighting with several designs. First, it proposes to conduct frame-wise relighting on the video diffusion model's paired VAE decoder. Besides, it proposes to progressively fuse the source video and relit video during the diffusion process to maintain temporal coherence. Further, a user-defined light trajectory injection module is applied. Finally, they use off-the-shelf normal estimation to provide the video's normal maps to enable a geometry-aware relighting module. Experiments demonstrate the effectiveness of the proposed approach.

**Strengths:**

- originality-wise: the idea of using a single-image relighting model to enable video relighting is interesting.
- quality-wise: the qualitative and quantitative results are promising.
- clarity-wise: the presentation is good.
- significance-wise: the video relighting is important for a lot of downstream tasks, e.g., content creation.

**Weaknesses:**

I feel the ablations in the paper are quite inadequate. Even though there are some ablations in the appendix that are good, they are not the core part of the model design.

For example, can authors provide both **quantitative and qualitative** results to show the gradual improvement from the pre-trained IC-Light? From my understanding, there are several enhancements, but I have no clue which one contributes. Feel free to add things that I miss.
- decoded frames + residual instead of raw frames (L216)
- progressive fusion in Eq. (2)
- geometry-aware feature in latent space
- geometry-aware in frequency space.

I am actually confused why not directly use the raw frames in the source video? Since the authors add back the difference between the raw videos and the decoded videos (L216), isn't this the same as just using the original video?

**Questions:**

See "weakness"

---

> ### Author Response · Authors · 2025-11-21
> **Reply-Part1**
>
> We thank the reviewer for the constructive comments. We provide our feedback as follows.
>
> **W1:More quantitative and qualitative results about the ablation study.**
>
> We appreciate the reviewer’s suggestion and agree that clarifying the contribution of each component would strengthen the paper. Our goal is to progressively build controllable video relighting ability starting from the pretrained IC-Light model, and each added module addresses a specific limitation that IC-Light alone cannot resolve. Below, we clarify the role of each component:
>
> ---
>
> **1. Ablation Study on Decoded frames + residual (L216)**
>
> This design compensates for the loss of local high-frequency information introduced by the VDM denoising process. The residual computed once at the first step restores these fine details while preserving the relighting effects progressively established through latent-space denoising. Using the raw frames directly would break this balance. As demonstrated in Fig. 10 in Appendix B (PDF Page 17), directly injecting raw frames causes the output video to inherit substantial illumination patterns and low-frequency structures from the input, which preserves appearance but ultimately fails to achieve correct relighting. In contrast, the residual formulation enables us to replenish structural details during the early denoising stages, while its influence gradually diminishes in later stages where the lighting appearance is formed. This yields the desired balance: retaining necessary fine-grained details without interfering with the generation of high-quality, trajectory-controlled relighting. We also compute AQ(Aesthetic Quality in V-Bench) and FVD for both, as shown in the following table (also in Appendix B Table 3(PDF Page 16),), further quantifying the improvement in video quality achieved by our design.
>
> | method             | AQ  $\uparrow$   | FVD $\downarrow$     |
> | ---------- | ------ | ------ |
> | Raw Frames | 0.5946 | 1037.4 |
> | Residual(Ours)   | 0.6114 | 993.1  |
>
>
> ---
>
> **2. Ablation Study on Progressive fusion in Eq. (2)**
>
> The progressive fusion mechanism is key to achieving temporal consistency. As illustrated in the left part of Fig. 11 (PDF Page 17), compared with the per-frame results produced by IC-Light, progressive fusion ensures that both the foreground and background remain noticeably coherent over time. In particular, the red-boxed background region shows that without progressive fusion, the background exhibits clear frame-to-frame fluctuations with no temporal stability, whereas our fusion strategy effectively suppresses such inconsistencies and maintains a stable appearance throughout the sequence.
>
> To objectively evaluate the improvement in temporal consistency brought by progressive fusion, we additionally report the average CLIP score across consecutive video frames in the following table (also in Appendix B Table 4 (PDF Page 16)), which reflects semantic and structural coherence over time. With progressive fusion, the model achieves higher CLIP scores, confirming its contribution to smoother and more stable video relighting.
>
> | method                  | CLIP-Score $\uparrow$|
> | ----------------------- | ---------- |
> | w/o Progressive fusion  | 0.9200     |
> | With Progressive fusion | 0.9634     |
>
> ---
>
> **3. Ablation Study on geometry-aware feature in latent space VS in frequency space.**
>
> Incorporating geometry cues directly in latent space by concatenating per-frame normals often leads to blurring artifacts and temporal instability, primarily because these independently estimated normals are not well aligned with the latent representations as they evolve during the denoising process. To overcome this limitation, we introduce a geometry-aware integration strategy in the frequency domain, which provides a more stable and illumination-invariant way of embedding geometric information.
> As illustrated in the right part of Appendix B Fig. 11(PDF Page 17), the frequency-domain formulation suppresses interference from the original scene lighting while better preserving the desired, trajectory-driven shading transitions. This enables the model to utilize geometric cues without being influenced by inconsistent low-frequency content carried over from the input video. As a result, our approach achieves a favorable balance—retaining the geometric detail contributed by normals while ensuring the temporal stability crucial for controllable video relighting.
>
> To further validate the benefit of this design, we evaluate video quality under both integration strategies. Consistent with the visual observations, the frequency-domain method delivers notably superior results across quantitative metrics, as shown in the Appendix B Table 2(PDF Page 15).
>
> | method             | AQ  $\uparrow$   | FVD $\downarrow$     |
> | ------------------ | ------ | ------ |
> | In latent space    | 0.5912 | 1078.3 |
> | in frequency space | 0.6114 | 993.1  |
>
> ---

---

> > ### Author Response · Authors · 2025-11-21
> > **Reply-Part2**
> >
> > **Q1:Confusion about the residual**
> >
> > We thank the reviewer for raising this important clarification point. As shown in the first ablation study in W1 (Fig. 10 in Appendix B(PDF Page 17)), although we reintroduce a residual computed from the difference between the raw source video and the decoded video, this does not make our system equivalent to directly using raw frames. Directly using raw frames introduces a large amount of low-frequency structures inherited from the original video, which strongly interfere with the relighting process and ultimately lead to failure of the intended illumination manipulation.
> > In our design, the residual $\Delta d$ is computed only once at the first step and remains fixed throughout the denoising process. At each subsequent step, we add it back with a gradually decreasing weight, as expressed below:
> >
> > $$I_t = Decode(\hat z_{0←t}) + \lambda_t \cdot \Delta d,\quad   \lambda_t = 1 - \frac{t}{T_m}.$$
> >
> > Here,$\hat z_{0←t}$denotes the noise-free component obtained by partially denoising the latent at step t.
> > This progressive residual injection achieves a desirable balance:
> > - Early in the process, $ \Delta d $ helps recover high-frequency details that would otherwise vanish through latent-space denoising;
> > - Later in the process, its influence naturally diminishes, allowing the model to focus on forming accurate, trajectory-controlled lighting appearance without inheriting the raw input’s illumination.
> >
> > Thus, the system is not equivalent to directly using raw frames; instead, it provides a controlled and principled mechanism for preserving structural detail while maintaining high-quality relighting.

---

### Comment · Area_Chair_Y6XW · 2025-11-26

Dear reviewers,

Please check the author's reply. Feel free to raise any questions or start a discussion, regardless of whether you will change the score.

Your AC.

---

### Meta-Review · Area_Chair_8kQM · 2026-01-07

**Summary:**

The paper proposes training-free controllable video relighting method by adapting the single image relighting to video with several design changes. All reviewers acknowledge that the method is novel, paper is well written and the results are solid.
The reviewers primarily questioned the necessity of user-defined trajectories and the completeness of ablation studies for specific modules like LMI and GAR.
Some concerns were raised regarding the physical realism of light in 3D-unaware environments and the use of simplistic evaluation metrics like $PSNR_{Y}$.
Reviewers also sought more rigorous documentation of the user study and broader evidence of the model’s generalizability across diverse video content.
Finally, multiple reviewers asked about the technical significance of the method given its reliance on existing backbones like IC-Light.

While some limitations remain such as lack of explicit 3D understanding for complex occlusions or limited eval metrics, this is a significant and practical first step for trajectory-driven video editing. The method's ability to achieve high-quality, temporally coherent relighting without expensive retraining or fine-tuning is of high interest to the community. ALso, most reviewers lean towards acceptance and agree that the method is novel. Hence, I vote for acceptance.

**Reviewer Concerns:**

The authors successfully resolved the ablation gaps by providing new quantitative data (FVD/AQ) and clarified the user study design and participant background. They also demonstrated framework generality by successfully adapting their method to the CogVideoX backbone and testing on complex, non-linear trajectories. However, the lack of true 3D awareness (e.g., handling occlusions or volumetric scattering) remains an inherent, acknowledged limitation of the current 2D approach. While the authors justified the use of small dataset size as community standards, evaluation on large eval sets still is missing.

**Reviewer Scores:**

Reviewer bAHR would likely increase to a 7, as their primary request for detailed quantitative and qualitative ablations was thoroughly addressed. Reviewer shr2 might would retain the score - theauthors provided specific evidence for trajectory utility and gave more details on user study, but the evidence that light trajectories are very useful are not too rigorous. Reviewer v3AC would likely maintain a 6, as they appreciated the CogVideoX results but noted that the fundamental 3D realism issues are still present. Reviewer fsKh would likely stay at 6 since the authors concerns on eval metric and eval set size still exists.

---

### Decision · Program_Chairs · 2026-01-26

Accept (Poster)